

# The role of serum and urinary biomarkers in the diagnosis of early diabetic nephropathy in patients with type 2 diabetes

Deyuan Zhang[1,2], Shandong Ye[1,3] and Tianrong Pan[2]

[1] School of Medicine, Shandong University, Jinan, Shandong, China
[2] Department of Endocrinology, The Second Affiliated Hospital of Anhui Medical University, Hefei, Anhui, China
[3] Department of Endocrinology, Anhui Provincial Hospital, Hefei, Anhui, China

## ABSTRACT

**Background:** Previous studies have shown that a variety of biomarkers are closely related to the occurrence and development of early-stage diabetic nephropathy (DN) in patients. The aim of this study was to evaluate the role of multiple sera and urinary biomarkers in the diagnosis of early-stage DN in patients with type 2 diabetes.

**Methods:** We enrolled 287 patients with type 2 diabetes, who were classified into normoalbuminuria ($n = 144$), microalbuminuria ($n = 94$), or macroalbuminuria ($n = 49$) groups based on their urine albumin to creatinine ratios (UACR), along with 42 healthy controls. We assessed 13 biomarkers, including transferrin (Tf), immunoglobulin G (IgG), podocalyxin, neutrophil gelatinase-associated lipocalin (NGAL), N-acetyl-beta-glucosaminidase, α-1-microglobulin, 8-hydroxy-deoxyguanosine, tumor necrosis factor-alpha (TNF-α), and interleukin-18 in urine samples, along with cystatin C, total bilirubin, and uric acid in sera samples, to evaluate their diagnostic roles. From the measurements, the blood neutrophil to lymphocyte ratio was also calculated.

**Results:** Urinary Tf, IgG, NGAL, and TNF-α were significantly related to the UACR. We calculated the area under the receiver operating characteristic curves (area under the curve) and found that urinary IgG (0.894), NGAL (0.875), Tf (0.861), TNF-α (0.763), and the combination of urinary Tf + IgG + TNF-α + NGAL (0.922) showed good diagnostic value for early-stage DN.

**Conclusions:** Urinary Tf, IgG, NGAL, TNF-α, and the combination of all four biomarkers demonstrated excellent diagnostic value for early-stage DN in patients with type 2 diabetes.

Corresponding author
Shandong Ye, yesd196406@163.com

## INTRODUCTION

Diabetic nephropathy (DN), also known as diabetic kidney disease, is a common and severe microvascular complication of type 2 diabetes that can result in end-stage kidney disease. With the increasing incidence of type 2 diabetes worldwide, DN has become a global health concern (*Wan, Xu & Dong, 2015*). The effective treatment of DN

requires glycemic control and antihypertensive measures (*Wan, Xu & Dong, 2015*). It is important to identify DN during its early stages as prompt treatment can reduce the medical and economic burden of this disease (*Tziomalos & Athyros, 2015*; *Gudehithlu et al., 2018*).

Currently, microalbuminuria is the most widely investigated biomarker for the diagnosis of DN. However, its diagnostic value in early-stage DN is limited as renal injury commonly precedes proteinuria. In a previous study, *Perkins et al. (2010)* demonstrated that only 52.2% of patients with advanced-stage DN presented with proteinuria during a 12-year follow-up study. In another study, only 33% of patients with type 1 diabetes having confirmed microalbuminuria developed clinically-apparent kidney disease (*Rossing, Hougaard & Parving, 2005*). Other studies have reaffirmed these findings, demonstrating that a considerable portion of diabetic patients with renal dysfunction do not have proteinuria (*An et al., 2009*; *Kramer et al., 2003*; *Zachwieja et al., 2010*). Therefore, there is a dire need for more sensitive and specific biomarkers for the diagnosis of early-stage DN.

It has been well-established that glomerular damage, tubular injury, inflammatory responses, and oxidative stress contribute to the development of diabetic kidney disease (*Gurley et al., 2018*). In this study, we evaluated 13 serum and urinary biomarkers involved in glomerular damage, tubular injury, inflammation, and oxidative stress, for their potential application in the diagnosis of early-stage DN.

## MATERIALS AND METHODS

### Patients

In this cross-sectional study, 287 patients with type 2 diabetes (according to the 1999 WHO criteria) who were hospitalized in the Department of Endocrinology, Second Affiliated Hospital of Anhui Medical University between January 2018 and March 2018, were selected. In the same period, 42 healthy medical volunteers from the Health Management Center of our hospital were selected as the healthy controls. The collection of specimens was completed by the nursing staff, and the samples were submitted to the clinical laboratory of our hospital for analysis. The 287 enrolled patients with type 2 diabetes who were classified as normoalbuminuric ($n = 144$), microalbuminuric ($n = 94$), and macroalbuminuric ($n = 49$) based on their urine albumin to creatinine ratios (UACR) of <30, 30–300, and >300 mg/g, respectively.

Patients with the following conditions were excluded from this study: severe cardiac, liver, and pancreatic diseases; primary glomerulonephritis or kidney diseases caused by secondary conditions other than diabetes; infection, malignancies, or autoimmune disease; and recent acute diabetic complications including ketoacidosis, hyperosmolar nonketotic diabetic coma, and lactic acidosis. In addition to the above diseases, the healthy controls were free of hypertension, hyperlipidemia, hyperuricemia, and hematological diseases. All patients enrolled in this study provided oral informed consent before the study was conducted. The research followed the tenets of the Declaration of Helsinki and was approved by the Medical Ethics Committee of Anhui Medical University (Ethical Application Ref: 2017038).

## Data collection

Demographic and clinical parameters, including gender, age, duration of diabetes, blood pressure, height, body weight, body mass index, and fundus lesions, were collected. Fasting blood samples were drawn, and hemoglobin A1c (HbA1c) was measured using the HA-8160 HbA1c analyzer (Arkray KDK, Kyoto, Japan). Fasting blood glucose, total cholesterol, triglyceride, low-density lipoprotein (LDL), total bilirubin (TBIL), serum creatinine, cystatin C (CysC), uric acid (UA), neutrophil count, and lymphocyte count were measured using UniCel Dxc 800 biochemical analyzer (Beckman Coulter, Brea, CA, USA). The neutrophil to lymphocyte ratio (NLR) and estimated glomerular filtration rate (eGFR) were calculated. The eGFR was calculated using the CKD-Epi formula (*Levey et al., 2009*). The first midstream urine in the morning was collected in a sterile cup and stored at −80 °C for analysis of urinary albumin, transferrin (Tf), N-acetyl-beta-glucosaminidase (NAG), immunoglobulin G (IgG), and α-1-microglobulin (α1MG) using an immunonephelometric assay with the BN2 analyzer (Siemens Healthcare Diagnostics, Deerfield, IL, USA). The picric acid method was used for determining urinary creatinine (Ucr) levels, while urinary podocalyxin (PCX), neutrophil gelatinase-associated lipocalin (NGAL), 8-hydroxy-deoxyguanosine (8-OHdG), tumor necrosis factor-alpha (TNF-α), and interleukin-18 (IL-18) were measured with a commercial enzyme-linked immunosorbent assay kits (Elabscience Biotechnology, Wuhan, Hubei, China).

All biomarker specimens were collected and tested at our hospital. In order to eliminate the effect of urine concentration or dilution on the results, all measurements from the urine were presented as the ratio of the measured values to Ucr. For values <$X$ (where $X$ is a number) in the raw data which were values that were below the detection threshold of our method used to measure biomarker concentrations, the threshold value were used as the estimate in the analysis.

## Statistical analysis

Data were analyzed using SPSS software version 16.0 (IBM, Chicago, IL, USA). Continuous variables with normal distribution were expressed as mean ± standard deviation, and non-normally distributed data were expressed as median (25th percentile, 75th percentile). Differences among the groups were analyzed by the one-way analysis of variance, and comparisons between two groups were analyzed by the Student's *t*-test. Categorical variables were compared using the Wilcoxon rank-sum test or chi-squared test. The multivariate linear regression analysis was used to analyze the correlation between the different biomarkers and UACR. Correlation with UACR was defined if the *t*-test for the partial regression coefficients were statistically significant in the regression models. Receiver operating characteristic curve (ROC) analysis were used to assess the diagnostic values of each biomarker and the combination of biomarkers, and also to test whether the null hypothesis that biomarker were not indicative of early-stage DN was true. The area under the curve (AUC), sensitivity, and specificity of parameters were calculated based on the ROC. The ROC curve was the plot of all possible pairs of sensitivity and specificity values obtained by scrutinizing all possible values of the cut off. The optimal cut off point with relevant sensitivity and specificity was found where the

Youden's index was maximum (*Lai, Tian & Schisterman, 2012*). *P* < 0.05 was considered statistically significant.

## RESULTS

The parameters of each group are shown in Table 1. The Tf, IgG, PCX, CysC, NAG, α1MG, 8-OHdG, IL-18, and UA levels of patients in the macroalbuminuria group were higher than those in the other three groups (*P* < 0.05 for UA, *P* < 0.001 for other parameters). The NGAL, TNF-α, and NLR levels in the macroalbuminuria group were higher than those of the healthy control and normoalbuminuria groups (*P* < 0.001). The NLR levels in the macroalbuminuria group were higher than those in the microalbuminuria group (*P* < 0.05). TBIL in the macroalbuminuria group was lower than the normoalbuminuria and microalbuminuria groups (*P* < 0.001). The Tf, IgG, PCX, NGAL, NAG, α1MG, 8-OHdG, TNF-α, IL-18, and NLR levels of patients in the microalbuminuria group were higher than the healthy controls and normoalbuminuria groups (*P* < 0.05 for NLR, *P* < 0.001 for other parameters). The Tf, PCX, NGAL, NAG, α1MG, TBIL, and IL-18 levels in patients with normoalbuminuria were higher than the healthy controls (*P* < 0.001 or *P* < 0.05).

The multivariate linear regression was performed for the biomarkers correlated with UACR, and adjustments were made to account for risk factors, such as gender, age, diabetes duration, HbA1c, eGFR, LDL, systolic blood pressure and diastolic blood pressure. Urinary Tf, IgG, NGAL, and TNF-α were found to be significantly related to the UACR in patients with type 2 diabetes (*P* < 0.001) (Table 2). The diagnostic value of urinary Tf, IgG, NGAL, and TNF-α were analyzed by ROC curves (Fig. 1) and the results showed that, in terms of AUC values, IgG > NGAL > Tf > TNF-α; in terms of sensitivity, NGAL > IgG > Tf > TNF-α; and in terms of specificity, IgG > Tf > TNF-α > NGAL (*P* < 0.001) (Table 3).

The diagnostic value of the combination of urinary Tf, IgG, NGAL, and TNF-α were analyzed with ROC curves (Fig. 2). The results showed that the combination of urinary Tf, IgG, NGAL, and TNF-α had the highest AUC when compared with the individual biomarkers (*P* < 0.001) (Table 3).

## DISCUSSION

Diabetic nephropathy is a chronic and progressive kidney disease characterized by microalbuminuria and decreased glomerular filtration rates, which can eventually lead to end-stage renal disease and death. However, its early diagnosis and intervention could allow for the prompt treatment of DN, which would delay the pathological progression of the disease. In this study, we evaluated several potential biomarkers for their diagnostic potential for detecting early-stage DN. The primary finding of this study was that urinary Tf, IgG, NGAL, and TNF-α levels, both individually and in combination, are valuable biomarkers for assessing early-stage DN in patients with type 2 diabetes.

While some highly sensitive and specific biomarkers were previously identified for the diagnosis of DN (*Hara et al., 2012*), this study covered multiple parameters involved in the
**Table 1  Clinical and laboratory characteristics of patients in the normoalbuminuria ($n = 144$), microalbuminuria ($n = 94$), macroalbuminuria ($n = 49$), and healthy control ($n = 42$) groups.**

| | Healthy controls ($n = 42$) | Normoalbuminuria ($n = 144$) | Microalbuminuria ($n = 94$) | Macroalbuminuria ($n = 49$) | $F/\chi^2$ | $p$-value |
|---|---|---|---|---|---|---|
| Gender | | | | | 0.189 | 0.979 |
| Male, $n$ (%) | 23 (54.80) | 83 (57.64) | 52 (55.32) | 28 (57.14) | | |
| Female, $n$ (%) | 19 (45.20) | 61 (42.36) | 42 (44.68) | 21 (42.86) | | |
| Age (years) | 54.33 ± 14.974 | 54.32 ± 14.07 | 55.49 ± 15.26 | 59.20 ± 13.03★★ | 3.674 | 0.013 |
| BMI (kg/m²) | 23.58 ± 3.44 | 25.51 ± 3.56 | 25.70 ± 4.32 | 25.45 ± 3.58 | 3.426 | 0.017 |
| Duration (years) | – | 5.00 (2.00, 10.00) | 7.1 (1.00, 13.00) | 10.00 (7.50, 19.00)**# | 18.904 | <0.001 |
| Diabetic retinopathy, $n$ (%) | – | 31 (21.53) | 32 (34.04)* | 35 (71.43)**## | 40.485 | <0.001 |
| Systolic blood pressure (mmHg) | 122.55 ± 10.31 | 128.72 ± 15.69★ | 136.66 ± 18.27★★** | 148.04 ± 21.34★★**## | 22.924 | <0.001 |
| Diastolic blood pressure (mmHg) | 73.02 ± 6.97 | 78.25 ± 9.76★ | 80.20 ± 10.54★★ | 81.96 ± 12.76★★* | 6.787 | <0.001 |
| FBG (mmol/L) | 4.89 (4.58, 5.31) | 7.45 (5.99, 9.73)★★ | 8.50 (6.69, 11.21)★★* | 9.09 (6.61, 11.16)★★* | 77.190 | 0.027 |
| Scr (μmol/L) | 57.50 (48.50, 77.00) | 71.00 (56.25, 85.00)★ | 70.00 (56.00, 87.50)★ | 105.00 (80.00, 149.00)★★**## | 55.409 | <0.001 |
| eGFR (mL/min/1.73 m²) | 153.91 ± 49.68 | 127.14 ± 41.00★★ | 122.97 ± 43.31★★ | 79.33 ± 44.53★★**## | 23.755 | <0.001 |
| TC (mmol/L) | 4.45 (3.64, 5.12) | 4.50 (3.92, 5.03) | 4.64 (4.02, 5.52) | 4.97 (4.09, 5.78)* | 6.634 | 0.085 |
| TG (mmol/L) | 1.16 (0.81, 1.89) | 1.38 (0.95, 2.12) | 1.49 (1.03, 2.65)★ | 1.72 (1.16, 2.66)★ | 9.907 | 0.019 |
| LDL (mmol/L) | 2.86 ± 1.03 | 2.82 ± 0.84 | 2.90 ± 0.81 | 3.06 ± 1.04 | 0.875 | 0.454 |
| HbA1c | 5.38 ± 0.44 | 8.93 ± 2.14★★ | 9.41 ± 2.50★★ | 8.92 ± 2.72★★ | 35.176 | <0.001 |
| Tf/Ucr (mg/g) | 1.72 (1.25, 2.48) | 2.18 (1.42, 3.02)★ | 5.79 (3.12, 8.77)★★** | 43.60 (21.40, 182.33)★★**## | 199.600 | <0.001 |
| IgG/Ucr (mg/g) | 4.44 (3.00, 6.59) | 5.23 (3.57, 6.61) | 12.85(8.55, 18.43)★★** | 106.19 (41.28, 374.07)★★**## | 178.440 | <0.001 |
| PCX/Ucr (μg/g) | 2.98(2.04, 5.09) | 4.60 (2.30, 7.20)★★ | 6.69 (3.16, 13.16)★★** | 9.93 (7.69, 15.53)★★**## | 72.919 | <0.001 |
| NGAL/Ucr (μg/g) | 13.03 (9.80, 23.93) | 20.15 (12.37, 31.35)★★ | 67.09 (38.39, 114.16)★★** | 60.44 (27.09, 106.00)★★** | 138.604 | <0.001 |
| CysC (mg/L) | 0.72 (0.61, 0.89) | 0.72 (0.59, 0.90) | 0.79 (0.60, 1.00) | 1.12 (0.81, 1.90)★★**## | 45.945 | <0.001 |
| NAG/Ucr (U/g) | 10.19 (6.52, 13.41) | 13.56 (8.24, 18.80)★★ | 25.26 (14.46, 42.28)★★** | 50.63 (30.24, 100.06)★★**## | 144.974 | <0.001 |
| α1MG/Ucr (mg/g) | 8.51(5.75, 14.78) | 11.88 (7.98, 18.46)★★ | 24.48 (13.92, 40.20)★★** | 77.56 (34.82, 121.76)★★**## | 122.952 | <0.001 |
| 8-OHdG/Ucr (μg/g) | 6.56 ± 3.50 | 7.36 ± 4.42 | 14.42 ± 10.47★★** | 32.46 ± 28.46★★**## | 51.849 | <0.001 |
| TBIL (mmol/L) | 10.35 (7.50, 13.48) | 12.00 (9.50, 15.48)★ | 11.65 (9.38, 14.63) | 9.20 (6.65, 10.95)*## | 25.074 | <0.001 |
| UA (μmol/L) | 290.38 ± 84.65 | 294.62 ± 84.24 | 293.80 ± 98.07 | 338.10 ± 91.55★*# | 3.441 | 0.017 |
| TNF-α/Ucr (ng/g) | 6.49 ± 2.78 | 8.09 ± 4.36 | 17.24 ± 11.02★★** | 15.34 ± 8.25★★** | 40.305 | <0.001 |
| IL-18/Ucr (ng/g) | 27.62 (18.13, 39.60) | 32.24 (20.91, 51.34)★ | 72.49 (39.56, 113.18)★★** | 93.28 (40.71, 141.29)★★**## | 89.449 | <0.001 |
| NLR | 1.39 (1.10, 2.17) | 1.53 (1.16, 2.09) | 1.92 (1.33, 2.83)★★ | 2.22 (1.71, 3.89)★★**# | 28.380 | <0.001 |

**Notes:**
★ $P < 0.05$.
★★ $P < 0.001$ vs. healthy controls.
\* $P < 0.05$.
** $P < 0.001$ vs. normoalbuminuria group.
\# $P < 0.05$.
## $P < 0.001$ vs. microalbuminuria group.
BMI, body mass index; FBG, fasting blood glucose; Scr, serum creatinine; eGFR, estimated glomerular filtration rate; TC, total cholesterol; TG, triglyceride; LDL, low-density lipoprotein; HbA1c, hemoglobin A1c; Tf, transferrin; IgG, immunoglobulin G; PCX, podocalyxin; NGAL, neutrophil gelatinase-associated lipocalin; CysC, cystatin C; NAG, N-acetyl-beta-glucosaminidase; α1MG, alpha-1-microglobulin; 8-OHdG, 8-hydroxy-deoxyguanosine; TBIL, total bilirubin; UA, uric acid; TNF-α, tumor necrosis factor-alpha; IL-18, interleukin-18; NLR, neutrophil to lymphocyte ratio; Ucr, urinary creatinine.

**Table 2 Correlation analysis between the biomarkers (Tf, IgG, NGAL, and TNF-α) and UACR in patients with type 2 diabetes.**

| | Partial regression coefficient (B) | Standard error (SE) | Standard partial regression coefficient (β) | t | p-value |
|---|---|---|---|---|---|
| **Tf** | | | | | |
| Unadjusted | 14.351 | 0.871 | 0.870 | 16.483 | <0.001 |
| Adjusted | 14.006 | 0.898 | 0.849 | 15.589 | <0.001 |
| **IgG** | | | | | |
| Unadjusted | 1.113 | 0.211 | 0.225 | 5.262 | <0.001 |
| Adjusted | 1.070 | 0.220 | 0.216 | 4.860 | <0.001 |
| **NGAL** | | | | | |
| Unadjusted | 0.126 | 0.032 | 0.158 | 3.970 | <0.001 |
| Adjusted | 0.127 | 0.032 | 0.158 | 3.943 | <0.001 |
| **TNF-α** | | | | | |
| Unadjusted | −2.090 | 0.295 | −0.356 | −7.086 | <0.001 |
| Adjusted | −1.939 | 0.308 | −0.330 | −6.294 | <0.001 |

**Note:**

Tf, transferrin; IgG, immunoglobulin G; NGAL, neutrophil gelatinase-associated lipocalin; TNF-α, tumor necrosis factor-alpha.

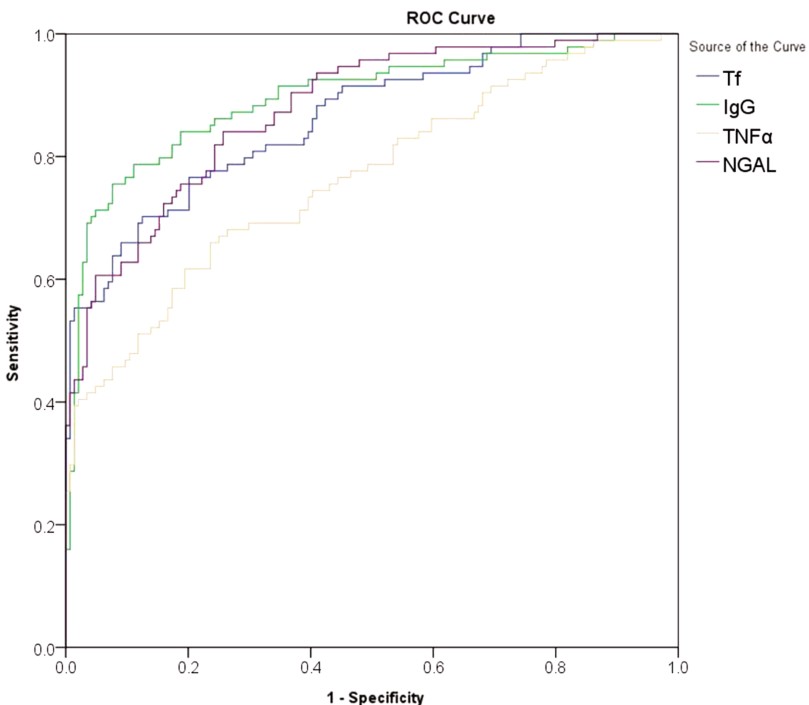

**Figure 1 ROC curves for the diagnosis of early-stage diabetic nephropathy using urinary Tf, IgG, NGAL, and TNF-α.** The abscissa represented specificity, the ordinate represented sensitivity, and the area under the curve represented the diagnostic value of urinary Tf, IgG,NGAL, and TNF-α in the early-stage diabetic nephropathy.

four major mechanisms of DN and evaluated the diagnostic value of them individually and in combination. The pathogenic mechanisms of DN include glomerular damage, tubular injury, inflammation, and oxidative stress (*Wada & Makino, 2013*; *Arora & Singh, 2013*).

**Table 3 Assessment of Tf, IgG, NGAL, and TNF-α in the diagnosis of early-stage diabetic nephropathy in patients with type 2 diabetes.**

| Parameters | AUC | 95% CI | Cut-off | Sensitivity (%) | Specificity (%) | p-value |
|---|---|---|---|---|---|---|
| Tf | 0.861 | 0.810–0.902 | 3.49 | 70.21 | 87.41 | <0.001 |
| IgG | 0.894 | 0.848–0.930 | 8.56 | 75.53 | 92.31 | <0.001 |
| NGAL | 0.875 | 0.826–0.915 | 30.03 | 84.04 | 74.13 | <0.001 |
| TNF-α | 0.763 | 0.704–0.816 | 10.46 | 65.96 | 76.22 | <0.001 |
| Combination | 0.922 | 0.880–0.953 | | 81.91 | 89.51 | <0.001 |
| UACR | 1.000 | 1.000 | 3.10 | 100.00 | 100.00 | <0.001 |

**Note:**

Tf, transferrin; IgG, immunoglobulin G; NGAL, neutrophil gelatinase-associated lipocalin; TNF-α, tumor necrosis factor alpha; UACR, urine albumin to creatinine ratios.

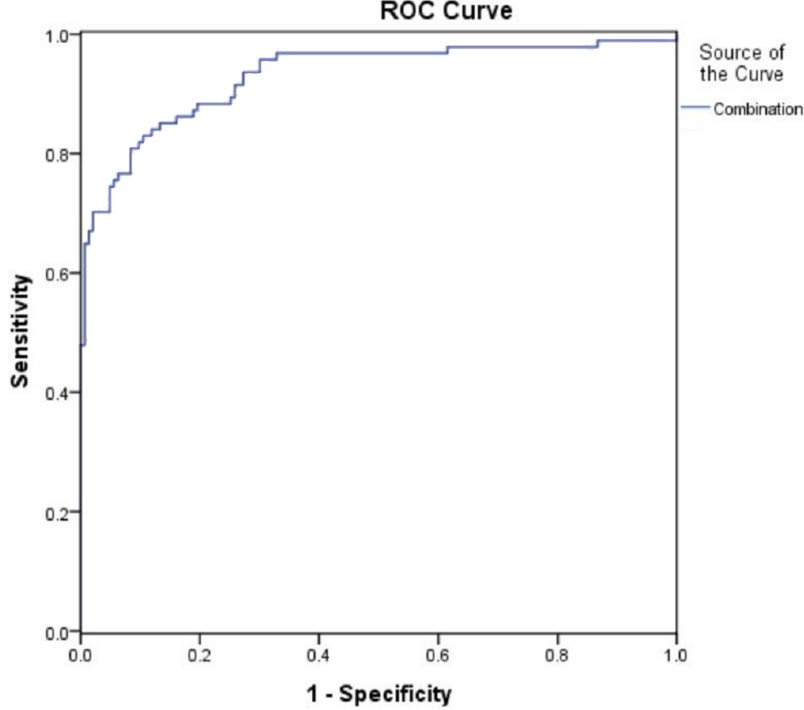

**Figure 2 ROC curves for the diagnosis of early-stage diabetic nephropathy using the combination of urinary Tf, IgG,TNF-α, and NGAL.** The abscissa represented specificity, the ordinate represented sensitivity, and the area under the curve represented the diagnostic value of the combination of urinary Tf, IgG,NGAL, and TNF-α in the early-stage diabetic nephropathy.

Therefore, biomarkers involved in these processes may potentially serve as clinically-relevant diagnostic parameters of DN (*Gluhovschi et al., 2016*).

Glomerular damage leads to increased urinary excretion of plasma proteins, such as Tf, IgG, and PCX (*Currie, McKay & Delles, 2014*). Tf is a plasma protein with a similar molecular weight as albumin. However, Tf is less anionic than albumin, which makes it more easily filtered via the glomerular barrier. Previous studies found that high urinary Tf levels in type 2 diabetic patients with and without microalbuminuria

(*Kanauchi, Akai & Hashimoto, 2002*; *Narita et al., 2004*). A 5-year follow-up study found that elevated urinary Tf levels could be used to predict the progression to microalbuminuria in patients with type 2 diabetes (*Narita et al., 2006*). While urinary IgG is also elevated in some patients prior to microalbuminuria (*Narita et al., 2004*). This suggests that urinary IgG levels may be a clinically-relevant biomarker for the early assessment and diagnosis of DN in patients with type 2 diabetes (*Narita et al., 2006*). In the current study, we found that urinary Tf and IgG may be used alone or in combination with other biomarkers for the diagnosis of early-stage DN. PCX, a negatively charged sialoglycoprotein, is expressed in podocytes and plays an essential role in maintaining the function of glomerular podocytes (*Lee & Choi, 2015*). Urinary PCX levels were found to be associated with UACR levels and were also increased in diabetic patients with normoalbuminuria (*Shoji et al., 2016*). While PCX may be an excellent marker for podocyte injury (*Hara et al., 2012*), it is not considered to be an excellent diagnostic parameter for DN by the current study.

Renal tubular injury is a critical characteristic of DN (*Kumar, Robertson & Burns, 2004*). Urinary tubular injury markers may increase in diabetic patients, even before the onset of microalbuminuria (*Russo et al., 2009*). NGAL, also named lipocalin-2, is a secretory protein that is released during renal injury and serves as a marker of acute kidney injury (*Bolignano et al., 2009*). CysC, an inhibitor of the cysteine protease, is freely filtered through the glomeruli and reabsorbed by the proximal tubule (*Jeon et al., 2011*). Previous studies demonstrated that urinary NGAL and CysC were higher in diabetic patients and positively associated with UACR (*Assal et al., 2013*; *Mahfouz, Assiri & Mukhtar, 2016*; *Rao et al., 2014*). Our study found that urinary NGAL was increased in diabetic patients with microalbuminuria, as well as diabetic patients with normoalbuminuria when compared with the healthy controls. However, CysC was only higher in patients with macroalbuminuria. Similarly, another study showed that CysC was only elevated in advanced-stage diabetic kidney disease and remained at normal levels during the early stages of the disease (*Ogawa et al., 2008*). NAG is a lysosomal enzyme expressed in proximal tubules and was previously found to be a sensitive marker of tubular injury (*Bazzi et al., 2002*). Urinary NAG was increased in type 2 diabetic patients compared to non-diabetic subjects, with higher levels in patients with microalbuminuria than those with normoalbuminuria levels (*Sheira et al., 2015*; *Bouvet et al., 2014*). The α1MG protein is freely filtered and reabsorbed through renal tubules. Urinary α1MG was also increased before the onset of microalbuminuria (*Shore, Khurshid & Saleem, 2010*; *Hong et al., 2003*). Although we failed to identify NAG and α1MG as early diagnostic markers, NAG was significantly elevated in the macroalbuminuria and microalbuminuria groups. Interestingly, α1MG levels differed among the four groups, suggesting that these two parameters might play a role in the assessment of disease severity and progression.

Oxidative stress is one of the mechanisms for DN (*Forbes, Coughlan & Cooper, 2008*). 8-OHdG is a stable oxidative product and a marker of DNA damage (*Valavanidis, Vlachogianni & Fiotakis, 2009*). Urinary 8-OHdG was increased in patients with type 2 diabetes and was found to be a marker for monitoring the progression of DN (*Waris et al., 2015*; *Hinokio et al., 2002*). However, another study found that urinary 8-OHdG

levels were similar in patients with DN and diabetic patients with normoalbuminuria (*Serdar et al., 2012*). In our study, although elevated urinary 8-OHdG levels were found in in the macroalbuminuria and microalbuminuria groups, urinary 8-OHdG was not considered to be a valuable diagnostic biomarker for early-stage DN. Bilirubin is an endogenous antioxidant, and low serum bilirubin levels are considered to be a risk factor for diabetic kidney damage (*Okada et al., 2014*). UA was reported to be associated with oxidative stress and endothelial dysfunction (*Jalal et al., 2011*), and was previously considered as a risk factor for DN (*Jalal et al., 2010*). However, another study found that UA levels were not associated with the progression of DN (*Ahola et al., 2017*). In the current study, TBIL and UA levels in the macroalbuminuria group differed from the normoalbuminuria and microalbuminuria groups, but there were no differences between the normoalbuminuria and microalbuminuria groups. Therefore, these two biomarkers were not considered as potential candidates for the detection of early-stage DN. However, they may be useful for assessing the severity of renal damage.

Diabetic nephropathy is a low-grade inflammatory disease (*Wada & Makino, 2013*), yet this study revealed a strong connection with TNF-α, which is a well-known inflammatory cytokine associated with renal injury (*Navarro et al., 2006*). Compared to the normoalbuminuria group, a 90% increase in urinary TNF-α excretion was present in type 2 diabetic patients with microalbuminuria, which was further correlated with the UACR (*Navarro et al., 2008*). Urinary IL-18 was positively associated with urine protein excretion and may be a predictive factor for assessing the progression of DN (*Liu et al., 2015*; *Nakamura et al., 2005*), yet one study disagreed with this finding (*Nadkarni et al., 2016*). NLR is a novel inflammatory marker known to be increased in patients with type 2 diabetes when compared with the healthy controls. NLR was also higher in patients with microalbuminuria group when compared with the normoalbuminuria group (*Huang et al., 2015*; *Khandare et al., 2017*). Our study found elevated levels of TNF-α, NLR, and IL-18 in patients with microalbuminuria and macroalbuminuria that were significantly higher than those detected in the healthy controls and type 2 diabetic patients with normoalbuminuria. However, only TNF-α was identified as a candidate biomarker for the detection of early-stage DN.

Using the ROC analysis, four biomarkers (urinary Tf, IgG, NGAL, and TNF-α) showed diagnostic value in terms of high AUC, in type 2 diabetic patients with DN. Urinary IgG showed a high specificity of 92.3%, yet the sensitivity was relatively low. Similarly, urinary NGAL was highly sensitive but suffered from its low specificity. Further analyses revealed the combination of these four biomarkers had better diagnostic value, in terms of AUC values, than those of the biomarkers individually. The AUC was as high as 0.922 for the combined biomarkers, indicating a superior value in the detection of early-stage DN.

## CONCLUSIONS

In conclusion, this study found that urinary Tf, IgG, NGAL, and TNF-α levels may be used as biomarkers for the diagnosis of early-stage DN in patients with type 2 diabetes. However, the combination of these four biomarkers showed higher sensitivity and specificity in comparison to the biomarkers when used individually. The four biomarkers assessed in this study are involved in glomerular damage, tubular injury, and

inflammation, respectively, which represents different aspects of the pathogenesis of DN. The combined use of these novel biomarkers may improve the detection of early-stage DN. However, further studies using larger sample sizes should be performed to establish the diagnostic value of other biomarkers for detection of early-stage DN.

## ACKNOWLEDGEMENTS

My deepest gratitude goes first and foremost to professor Shandong Ye.

### Funding

The research was supported by the Natural Science Foundation (no. 1808085QH292) of Anhui province of China and the local Scientific and Technological Development Project guided by the central government of China (no. 2017070802D147). The funders had no role in study design, data collection and analysis, decision to publish, or preparation of the manuscript.

### Grant Disclosures

The following grant information was disclosed by the authors:
Natural Science Foundation: 1808085QH292.
Local Scientific and Technological Development project guided by the central government of China: 2017070802D147.

### Competing Interests

The authors declare that they have no competing interests.

### Author Contributions

- Deyuan Zhang conceived and designed the experiments, performed the experiments, analyzed the data, contributed reagents/materials/analysis tools, prepared figures and/or tables, authored or reviewed drafts of the paper, approved the final draft.
- Shandong Ye conceived and designed the experiments, contributed reagents/materials/analysis tools, authored or reviewed drafts of the paper, approved the final draft.
- Tianrong Pan approved the final draft.

### Human Ethics

The following information was supplied relating to ethical approvals (i.e., approving body and any reference numbers):

Medical Ethics Committee of Anhui Medical University granted Ethical approval to carry out the study within its facilities (Ethical Application Ref: 2017038).

### Data Availability

The raw measurements are available in the Supplemental Files. The raw data shows all healthy controls and type 2 diabetes with or without proteinuria. These data were used for statistical analysis comparisons.

## Supplemental Information

Supplemental information for this article can be found online at http://dx.doi.org/10.7717/peerj.7079#supplemental-information.

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
