# Peer review of "The role of serum and urinary biomarkers in the diagnosis of early diabetic nephropathy in patients with type 2 diabetes"

_PeerJ, doi:10.7717/peerj.7079_

## Round 0.1 · original submission · Major Revisions

Dear Drs. Yuan and Dong,

Your manuscript entitled " The role of serum and urinary biomarkers in the diagnosis of early diabetic nephropathy in patients with type 2 diabetes" which you submitted to PeerJ, has been reviewed by the editor and 2 experts in the field.

I regret to inform you that the reviewers have raised serious concerns, and therefore your paper cannot be accepted for publication in PeerJ in its present form. However, since reviewers felt the manuscript contained some potentially interesting data, I would be willing to reconsider if you wish to undertake major revisions and resubmit.

If you decide to resubmit the revised version, please summarize all the improvements made in the new version and give answers to all critical points raised in the reviewers’ report in an accompanying letter. Please copy and paste each and every reviewer's comment above your response. If you feel any of their points are inappropriate, you are certainly free to provide rebuttal in your covering letter.

I strongly suggest improving the statistical analysis according to the reviewers' suggestions and providing a more detailed description of the method section.

Please note that resubmitting your manuscript does not guarantee eventual acceptance. Since the requested changes are major, the revised manuscript will undergo a second round of review by the same reviewers. I must emphasize that the acceptability of the revision will depend upon the resolution of the points raised by the reviewers.

Sincerely yours,

Stefano Menini

Reviewer 1 ·

Basic reporting

• The English used in the paper is in general clear and unambiguous, but with some less clear expressions.

• The article has a sufficient introduction where the reason for further examination of biomarkers is explained with relevant references. The abstract does however lack a sentence or two of the background of the study. The text written under ‘Background’ is more the aim of the study. Please add ‘The aim of’ in front of This study in line 23.

• There is however some less clear definition of diabetic nephropathy as it seems as microalbuminuria is defined as diagnostic of diabetic nephropathy. Diabetic nephropathy is defined in both type 1 and type 2 diabetes as the presence of persisting severely elevated albuminuria of more than 300 mg/24 hour (or > 200 µg/min) or an albumin creatinine ratio >300 mg/g creatinine, confirmed in at least two out of three samples, with concurrent presence of diabetic retinopathy and absence of signs of other forms of renal disease (Parving H-H et al. Diabetic Nephropathy. In: al. MWTe, editor. Brenner and Rector: The KIDNEY. 1. Philadelphia, USA: Elsevier; 2012. p. 1411-54.)

• The structure of the article conforms to an acceptable format which also counts the figures and tables. Raw data is available.

Experimental design

Re. Research question
• The research question is as stated, if there is a role of multiple serum and urinary biomarkers in the early diagnosis of diabetic nephropathy. But it is not clear how “early diagnosis of diabetic nephropathy” is explained. Please add that. Many of the biomarkers have been examined in diabetic cohorts previously, why it would be relevant to include a description of the novelty of this study or knowledge gap being investigated and how this study contributes to filling that gap.

• The study has been conducted in accordance with the prevailing ethical standards in the field.

Re. Methods
Patients:
• The ‘Patients’ section: please add ‘cross-sectional’ after ‘In this’ line 65.
• When was the study conducted? (year), how was type 2 diabetes defined (according to WHO criteria?), how was ‘healthy controls’ defined? From where were the patients recruited? (outpatient clinic?, general practice? Hospital?) and the healthy controls? Please add more information on details about how the participants were examined (how many days, where and by whom nurse, doctor, as part of a visit in the clinic etc) how demographic data was collected and from where.
Data collection:
• Re. urine: was the UACR based on a first voided morning urine? Please add. Was the characterization of albuminuria group based on actual albuminuria (on the examination day) or historical albuminuria? Was the characterization based on a single or several UACR measurements? How was UACR measured?
• For all biomarkers: please add the length of the period from sample collection to analysis and please add how the samples were stored in the meantime (at minus 80 degrees C etc)
• Re.eGFR: based on which method was that calculated (CKD-Epi?)
Statistical analysis:
• Quite often biomarkers are non-normally distributed, please add information on this and if skewed data, then information on log-transformation
• Line 98 and 99: Instead of compression, I suggest the use of ‘comparisons’
• Please explain why logistic regression analysis has been used instead of linear regression?
• There is no information of adjustments for traditional risk factors: gender, Hba1c, eGFR, smoking, LDL, systolic BP and other conventional risk factors. Adjustment is highly recommended to give data clinical relevance.
• ROC curves and the AUC are widely used to examine the effectiveness of diagnostic markers but can be strongly influenced by covariate variables. Please include relevant covariates in the analysis.
• To account for the influence of urinary dilution of the biomarker concentration, please normalize urine biomarkers to the urine creatine concentration.

Validity of the findings

• Impact and novelty not firmly assessed.
• The recommended statistical analyses need to be performed before validity of findings can be evaluated.
• In the introduction it is stated that there is “a dire need for more sensitive and specific biomarkers for the early diagnosis of diabetic nephropathy” (line 55-56) since “its (UACR) early diagnostic value in DN is limited as renal injury commonly precedes proteinuria”. It therefore seems contradictive to write in the conclusion (line 223-224) “The combined use of these novel biomarkers, along with the traditional marker (microalbuminuria) may improve the early detection of diabetic nephropathy”. Please clarify.

Additional comments

This manuscript by Yuan et al. examined the associations between 13 biomarkers and the early diagnosis of nephropathy in 287 patients with T2D and various degrees of albuminuria and a control group of healthy individuals. The topic of the paper is relevant, and I commend the authors for an impressing number of participants included. There are some major issues in the methodology, including the statistical analysis which should be improved as well as a more detailed description of the method section.

·

Basic reporting

The article is written in clear and professional English. However, there are some parts where the language and the used terminology could be improved:

1. The description of the calculation of AUC, sensitivity and specificity based ROC analysis on lines 101-103 should be improved.
2. It is not clear to which analysis are lines 103-106 referring to since the same has been said already before on the “Statistical analysis” section.
3. The meaning of the term “compression” on lines 90, 102 is not clear.
4. Term “normoalbuminuria” is more commonly used than “normal proteinuria”.
5. The correct spelling of “Wilcoxon rank sum” is “Wilcoxon rank-sum” and “chi-square test” is “chi-squared test”

The authors provide sufficient background on the importance for the early detection of diabetic nephropathy (DN), the short-comings of albuminuria in predicting development of DN and the need for improvement in the early detection of DN.

The article follows the acceptable format of ‘standard sections’ and figures and tables are relevant to the content of the article. The raw data has been made available as excel-files.

Experimental design

The article represent original primary research within aims and scope of the journal. The research has been conducted rigorously and to a high technical and ethical standard. The research question is well formulated but the design of the experiments does not support answering the identified knowledge gap:

1. In the introduction it is stated that 13 biomarkers are evaluated for their potential application in the prediction of DN. It is not possible to evaluate the predictive value in a cross-sectional study setting. Some follow-up data on events related DN should be obtained for this purpose.
2. It is unclear how the identified urinary biomarkers Tf, NGAL and TNF-alpha help in the early diagnosis of DN compared to the gold standard UACR. It is possible that they would be useful on top of UACR in predicting onset of DN, but to test this hypothesis, follow-up data would be needed.
3. It is not clear what is the utility of determining biomarkers that are correlated with microalbuminuria which itself is a biomarker of DN.

Methods have been described in sufficient detail but there are some parts that could be clarified:

1. It is not clear how the sensitivity and specificity in Table 3 and Table 4 have been calculated. I assume they have been evaluated at the ‘Cut-off’ but this should be clearly stated as well as how ‘Cut-off’ was defined.
2. It should be stated how eGFR was calculated. Was it done with CKD-EPI equation or with some other equation?
3. It should be stated for which test the p-value is given in Table 3 and Table 4.

Validity of the findings

The data is robust and statistically sound. However the are some issues:

1. The urinary biomarkers found to be associated with microalbuminuria, namely, Tf, NGAL and TNF-alpha, have been previously shown to be associated with UACR in T2D individuals as noted by the authors. It should be clearly stated what is the added value to the literature of investigating this associations again.

Additional comments

I congratulate the authors for their well phenotyped dataset and the manuscript which is well written in professional language. The biggest weakness is that the identified knowledge gap, prediction of DN cannot be achieved with the design of the experiment. Follow-up data is needed to infer possible utility of the biomarkers in predicting onset of DN. In addition, as the authors note, urinary Tf, NGAL and TNF-alpha have been previously shown to be associated with UACR in T2D individuals. It should be clearly stated what is the added value to the literature of replicating these findings in the presented manuscript.

Reviewer 3 ·

Basic reporting

see comments

Experimental design

see comments

Validity of the findings

see comments

Additional comments

In this work, Yan et al. analyze plasma and urine biomarkers in a cohort of diabetic patients carachterized by different degrees of proteinuria.
They found that a number of urine biomarkers were increased in parallel with the increase of proteinuria. They also calculated risk with a multivariate analysis between patients with normal proteinuria and those with microalbuminuria and also estimated the diagnostic value of a few urine biomarkers (NGAL, Tf, TNF-alpha) using a ROC curve. They concluded that these biomarkers, alone or in combination, could represent early diagnostic biomarkers od diabetic nephropathy.

The work is well written, concise and of broad potencial interest for the nephrologists community.
The background is well framed. However there are some assumption and lacking controls that weaken the conclusions of the work.


Major.
- As they also stated in the introduction the presence of albuminuria is not an essential condition associated with DN. Only kidney biopsy allows an etiological diagnosis of kidney damage and DN. In this work there is no histologically proven diagnosis of DN. As a consequence, the biomarkers investigated in this study can only be correlated with the degree of proteinuria and not with early or late stage of DN. In other words, microalbuminuria cannot be considered tout court a proven sign of early diabetic nephropathy.
- The hypothesis that transferrin could be a potential diagnosis biomarker of DN is not novel (Jpn J Nephrol 37:649-654, 1995), as it is that of NGAL (Diabet. Med. 27, 1144–1150 (2010) , TNF- (NDT (2006) 21: 3428–3434).
- In most of the studies conducted on potential novel urine biomarker for kidney diseases, their amount is normalized for the urine creatinine (Ucr), considering that the degree of urine output is extremely variable depending on water intake, room temperature, exercise… In this work , the authors did non perform a normalization of their data. Authors should normalize their data before performing the downstream analysis.
Minor:
- Authors should better describe the results reported in Table 2 and explicits the labels (, Wald, OR) to allow an easier understanding of the data.

---

## Round 0.2 · Minor Revisions

Dear Drs. Zhang and Ye,

Your manuscript entitled " The role of serum and urinary biomarkers in the diagnosis of early diabetic nephropathy in patients with type 2 diabetes" which you submitted to PeerJ, has been re-reviewed. The reviewer comments are included at the bottom of this letter and in the attached document.

The reviews are in general favourable and suggest that, subject to minor revisions, your paper could be suitable for publication. I would urge you to give these points your careful attention.

I hope that you will be prepared to make the necessary amendments and submit a revised manuscript accompanied by a statement of how you have responded to the criticisms raised.

I look forward to receiving your revision.

Sincerely yours,

Stefano Menini

Reviewer 1 ·

Basic reporting

Nothing further to add

Experimental design

Nothing further to add

Validity of the findings

Nothing further to add

Annotated reviews are not available for download in order to protect the identity of reviewers who chose to remain anonymous.

·

Basic reporting

The article is written in clear and professional English. However, there are some parts where the language could be improved as well as some small typographical errors:
1. On lines 69-70 I suggest using normoalbuminuric, microalbuminuric and macroalbuminuric.
2. On lines 108-109 the language is not clear. I suggest, “Differences among the groups were analyzed …”.
3. Table 1 uses “normal proteinuria” instead of “normoalbuminuria” used otherwise in the manuscript.
4. First paragraph of the results lines 118-129: here P<0.01 is used but according to Table 1 it should be P<0.001. Which one is correct?
5. Line 163: Instead of “the progression of microalbuminuria”, I suggest “the progression to microalbuminuria” or “the development of microalbuminuria”
6. Table3. The 95% CI for Tf does not include the reported AUC value?

The authors provide sufficient background on the importance for the diagnosis of early-stage diabetic nephropathy (DN), the short-comings of albuminuria in predicting development of DN and the need for improvement in the early detection of DN.

The article follows the acceptable format of ‘standard sections’ and figures and tables are relevant to the content of the article. The raw data has been made available as excel-files.

Experimental design

The article represent original primary research within aims and scope of the journal. The research has been conducted rigorously and to a high technical and ethical standard. The research question is well formulated and the knowledge gap being investigated is identified. However methods section needs more detail:
1. Lines 130-131: How were the biomarkers shown in Table 2 chosen for analysis? Please clearly define what was considered correlation with UACR.
2. Line 114: Please describe how sensitivity and specificity were calculated more precisely. For example “Sensitivity and specificity were evaluated at a cut-off value which maximized Youden’s index”. Assuming that Youden’s index was used?
3. It should be stated for which statistical model and null hypothesis are the p-values in Table 3 reported to.
4. The raw biomarker data contains values <X (where X is a number). I assume these are values that were below the detection threshold of the method used to measure biomarker concentrations. It should be stated how these values were treated in the analysis.

Validity of the findings

The data is robust and statistically sound. However the are some issues:
1. Please discuss why the ‘Partial regression coefficient of TNF-alpha in Table 2 is negative, whereas in Table 1 the TNF-alpha seems to increase from normoalbuminuria group to microalbuminuria and macroalbuminuria group?
2. Please discuss why the ROC analysis is not adjusted for covariates.

Additional comments

I congratulate the authors for their well phenotyped dataset and the manuscript which is well written in professional language. The presented data is interesting but the methods section still needs more detail.

---

## Round 0.3 · Minor Revisions

Dear Drs. Yuan and Dong,

Your manuscript entitled "The role of serum and urinary biomarkers in the diagnosis of early diabetic nephropathy in patients with type 2 diabetes" has again been carefully reviewed by the Editor and Reviewer 2. Basically the revision is now acceptable for publication, but before final acceptance is given, I would appreciate it if you would address the remaining minor comments of Reviewer 2.

Please summarize all the improvements made in the new version and give answers to all critical points raised in the reviewer’s report in an accompanying letter. Please copy and paste each and every reviewer's comment above your response.

If you are willing to do this, it would not be necessary for me to return the manuscript to the reviewers, but it could then be accepted for publication.

Sincerely yours,

Stefano Menini

·

Basic reporting

For lines 69-70 I would use the form albuminuric instead of albuminuria since you are talking about patients classified into these groups.

The footnote of the Table 1. (line 411) uses "normal proteinuria" instead of "normoalbuminuria" used elsewhere in the manuscript. I suggest using consistently one term.

On line 123 I would suggest removing "Youden’s index was calculated by “sensitivity and specificity minus 1” and adding reference for Youden's index in the previous sentence.

Experimental design

Please clarify which statistical test was used to test the hypothesis that the biomarkers were not indicative of early-stage DN (lines 118-119).

Validity of the findings

no comments

Additional comments

Thank you for answering my questions and addressing my comments. I have only few minor comments attached here.

---

## Round 0.4 · accepted · Accept

Dear Drs. Zhang and Ye,

Thank you for submitting a revised version of your manuscript entitled "The role of serum and urinary biomarkers in the diagnosis of early diabetic nephropathy in patients with type 2 diabetes".

I am pleased to inform you that your manuscript is accepted for publication in PeerJ in its current form and will now be forwarded to the product editor for copy editing and publication.

I thank the reviewers for their effort in improving the manuscript and the authors for their cooperation throughout the review process

Yours sincerely,

Stefano Menini